# Microneedle-Assisted Transfersomes as a Transdermal Delivery System for Aspirin

**DOI:** 10.3390/pharmaceutics16010057

**Published:** 2023-12-29

**Authors:** Raha Rahbari, Lewis Francis, Owen J. Guy, Sanjiv Sharma, Christopher Von Ruhland, Zhidao Xia

**Affiliations:** 1Centre for Nanohealth, Institute of Life Science 2, Swansea University Medical School, Swansea SA2 8PP, UK; 2Department of Chemistry, School of Engineering and Applied Sciences, Faculty of Science and Engineering, Swansea University, Swansea SA2 8PP, UK; o.j.guy@swansea.ac.uk; 3Department of Biomedical Engineering, School of Engineering and Applied Sciences, Faculty of Science and Engineering, Swansea University, Swansea SA2 8PP, UK; 4Electron Microscopy Unit, Central Biotechnology Services, Institute for Translation, Innovation, Methodology and Engagement, School of Medicine, Cardiff University, Cardiff CF14 4XN, UK; vonruhlandcj@cardiff.ac.uk

**Keywords:** microneedle, transfersome, transdermal delivery, aspirin

## Abstract

Transdermal drug delivery systems offer several advantages over conventional oral or hypodermic administration due to the avoidance of first-pass drug metabolism and gastrointestinal degradation as well as patients’ convenience due to a minimally invasive and painless approach. A novel transdermal drug delivery system, comprising a combination of transfersomes with either solid silicon or solid polycarbonate microneedles has been developed for the transdermal delivery of aspirin. Aspirin was encapsulated inside transfersomes using a “thin-film hydration sonication” technique, yielding an encapsulation efficiency of approximately 67.5%. The fabricated transfersomes have been optimised and fully characterised in terms of average size distribution and uniformity, surface charge and stability (shelf-life). Transdermal delivery, enhanced by microneedle penetration, allows the superior permeation of transfersomes into perforated porcine skin and has been extensively characterised using optical coherence tomography (OCT) and transmission electron microscopy (TEM). In vitro permeation studies revealed that transfersomes enhanced the permeability of aspirin by more than four times in comparison to the delivery of unencapsulated “free” aspirin. The microneedle-assisted delivery of transfersomes encapsulating aspirin yielded 13-fold and 10-fold increases in permeation using silicon and polycarbonate microneedles, respectively, in comparison with delivery using only transfersomes. The cytotoxicity of different dose regimens of transfersomes encapsulating aspirin showed that encapsulated aspirin became cytotoxic at concentrations of ≥100 μg/mL. The results presented demonstrate that the transfersomes could resolve the solubility issues of low-water-soluble drugs and enable their slow and controlled release. Microneedles enhance the delivery of transfersomes into deeper skin layers, providing a very effective system for the systemic delivery of drugs. This combined drug delivery system can potentially be utilised for numerous drug treatments.

## 1. Introduction

Transdermal drug delivery is a highly promising alternative method to the oral and hypodermic administrations of drugs. The transdermal administration of drugs can overcome the problems associated with the absorption and degradation of drugs in the gastrointestinal tract and liver, reducing drug side effects and cytotoxicity, whilst increasing the absorption and bioavailability of drugs and improving patient compliance [1,2,3]. However, the *stratum corneum* imposes significant restrictions on the delivery of molecules and drugs across the skin [4,5]. Thus, numerous transdermal drug delivery techniques have been developed to circumvent this barrier. These methods include chemical penetration enhancer and physical and electrical enhancers such as thermal ablation, electroporation and sonophoresis and topical patches [6,7].

The encapsulation of drugs inside polymeric and lipid-based particles provides a mechanism for the transdermal delivery of drugs and their controlled release. These drug encapsulation methods have been studied for both topical and transdermal delivery [8,9,10].

A controlled drug delivery system can control dose and drug release at a predetermined rate [11,12,13]. One of the main advantages of the encapsulation of drugs inside vesicle/nanoparticles is to improve the solubility of drugs and maximise the amount of a poorly water-soluble drug contained in a small, concentrated volume—by entrapment inside the vesicles [14]. Thus, when encapsulated, larger quantities of the drug can be applied onto the skin relative to the case of the free drug—which would require a large volume of drug solution to deliver a similar drug quantity to the skin surface.

There are different types of phospholipid-based vesicles used to encapsulate drugs for transdermal and topical drug delivery. These include liposomes, micelles, transfersomes and other nanostructures [15,16]. Among all lipid-based vesicles for drug encapsulation, transfersomes (TFs) have an advantageous elasticity due to the incorporation of an edge activator (surfactant) in the lipid bilayer, which yields improved skin permeability resolving the problem associated with the inflexibility of liposomes [17,18,19,20]. The term “transfersome” was initially introduced by Cevc and Blume in 1992 [19]. Since then, both terms “transfersomes” and “transferosomes” have been employed interchangeably to denote ultradeformable vesicles. TFs have excellent biocompatibility and are able to deform themselves to pass through very narrow constrictive channels that are up to 10 times smaller than their own diameter without experiencing a measurable loss of the encapsulated drug [21,22]. TFs penetrate through the *stratum corneum* lipid lamellar regions as a result of the osmotic forces in the skin, which drive diffusion from the low hydration surface of the skin into the highly hydrated aqueous skin regions [23,24].

It has been suggested that there are two possible modes of action of TF as a transdermal drug delivery system. First, the TF can act as a drug carrier system, diffuse into the stratum corneum, and carry the encapsulated drug into the skin layer. Secondly, the TF can act as a permeation enhancer, which enters the stratum corneum, modifying the intercellular lipid structure (as it can fuse and bind to the lipid-bilayer structure of the skin) and release the drug. This enhances the penetration of the released drug molecules, which are then able to be transported via intercellular and intracellular pathways across the skin layers. Both modes of action of TFs can occur at the same time [25,26], but depending on the physico-chemical properties of the drug molecules (partition coefficient, molecular size, solubility, melting point and ionisations), one of the mechanisms may be more dominant than the other [27]. The release mechanism from phospholipid TFs is dependent on the surrounding environment and on the nature of the encapsulated drug (hydrophilic/hydrophobic). For instance, a hydrophilic drug would be expected to leak out of the TF into the surrounding aqueous environment at a higher rate than a corresponding hydrophobic drug [28]. In both cases, the release mechanism occurs through the leakage of the drug through the flexible and mobile phospholipid membrane. TFs are an effective and beneficial drug encapsulation technique, balancing the hydrophobicity and hydrophilicity of drugs for their more efficient diffusion across both dry and hydrated skin layers and the controlled release of drug. The different types of interaction of TF with the cells depend on the cell type, the phospholipid composition of TF, and the presence of specific receptors on the surfaces of the cells [29,30].

Microneedles are another method which have garnered considerable attention in transdermal delivery research as an effective method for the pain-free delivery of drugs [31,32,33].

Microneedles consist of micro-projection arrays of microneedles on a backing substrate, which pierce the *stratum corneum*, thus bypassing this barrier layer. Although the mechanism of the effect of microneedles is based on skin pore creation, they do not cause skin damage as the generated pore channels quickly close due to the elastic retraction force of the skin [34]. Microneedles are very effective for the delivery of drugs and molecules in a minimally invasive and pain-free manner [35,36]. There are several types of microneedles that can be used in the microneedle-assisted TF delivery system. These include solid microneedles, hollow microneedles, coated microneedles, and dissolvable microneedles, which can be made from polymer, metal, silicon, or glass and ceramics, with different insertion sizes and shapes [37,38,39]. In this study, solid silicon and solid polycarbonate microneedles were utilised. Solid microneedles can be used as a skin pre-treatment, forming micro-scale holes or pores in the superficial skin layers [27,35] for a subsequently enhanced delivery. After the insertion of the solid microneedles, the drug formulation can be applied over the micro-pores on the skin surface, which then facilitates the diffusion of the drugs and the molecules through the pores across the skin layers into the body, for both a local and systemic effect [40].

A combined system of microneedles and TF could provide a solution for faster permeation and release into deeper skin layers for an effective systemic transdermal drug delivery [40,41,42,43]. Microneedles enable the TF-encapsulated (TF-Asp) drug to surpass the stratum corneum, and subsequently release the drug over a sustained period of time with the flexibility of TF helping TF diffuse into deeper skin layers for a more effective system delivery. It should be noted that, for certain skin conditions, where the targeted delivery is mainly required to deliver the drug into the epidermal skin layer (e.g., skin conditions including psoriasis and eczema), TF application is more favourable than the dual-microneedles/TF approach.

Numerous research studies have combined microneedles and a nano-carrier drug delivery system. The study by Jing et al. developed the microneedle-assisted delivery of HaCaT cell membrane-coated pH sensitive nano-carriers (micelles). The micelles were internalised and mainly accumulated in the active epidermal layer for the treatment of psoriasis [44]. In another study, Wu et al. developed dissolvable microneedles to deliver nano-carriers to membrane-coated hypertrophic scar fibroblasts for the treatment of hypertrophic scars [45].

Xuanjin et al. developed dissolvable microneedles integrated with oppositely surface-charged transfersome-encapsulating antigen to investigate their effect on the immune response via transdermal immunisation [46].

In this work, a combined microneedle/TF transdermal delivery platform, that can be used for the delivery of a number of different drugs, is presented. The combination of microneedles (for enhanced permeation through the micro-pores created in the skin by the microneedle) and TFs (allowing the encapsulation of drugs and controlled release following delivery into the skin) is shown to be significantly more effective than either individual method.

The microneedle–TF combination delivery technique has been developed using aspirin as the active test drug. Aspirin has universal benefits (anti-inflammatory, reducing the risk of cardio-vascular disease and effects in different types of cancer [47,48,49,50]). It has also been shown to be beneficial in various dermatological disorders including Hughes’ syndrome, mild type 1 lepra reaction, and melanoma [51]. However, aspirin can cause several side effects via oral administration such as stomach ulcers. The development of a transdermal method for aspirin delivery could have significant therapeutic benefits for a number of diseases and conditions, such as a local painkiller for osteoarthritis; stroke prevention; and the treatment of different type of cancers. TF compositions, modified by changing the proportion of surfactant (Tween 80) in the TF formulation, were optimised as previously reported [52]. The composition can be used to modulate the aspirin release rates from the TFs. In this study, the optimised form of TF–Asp has been utilised for further characterisation, drug release and permeation studies—with and without the microneedle pre-treatment of porcine skin.

## 2. Materials and Methods

### 2.1. Materials

Chemicals including HPLC grade methanol (34860-2.5L-R), acetic acid (45754-500ML-F), HPLC grade water (270733-2.5L), acetyl salicylic acid (A5376-100G), egg yolk L-α-phosphotidycholine, lypophilised powder (P3556-1G), and Tween-80 viscous liquid (P1754-500ML) were supplied by Sigma Aldrich, Gillingham, UK. Dialysis sack (MW cut of 12000) (D9777-100FT) was obtained from Sigma Aldrich, Gillingham, UK. DMEM-F12 (31331-028) foetal bovine serum (10100147-500ML), penicillin–streptomycin (15070063, 100ML), 1.5 Mm glutamine (A2916801-200ML), phosphate-buffered saline (PBS) (10010-500ML), MEM phenol-free (51200038-500ML), and 1 mM cell-labelling solution Vybrant Dil (V22885) were supplied by Thermo Fisher Scientific, Waltham, MA, USA. AlamarBlue (BUF012B) was obtained from Bio-Rad, Hercules, CA, USA. Human skin fibroblasts (primary cells originated from a patient) were kindly supplied by the Department of Medicine at Swansea University. Au nanoparticles (Au-NPs) were provided by the School of Engineering at Swansea University.

### 2.2. Methodology

#### 2.2.1. Fabrication of Transfersomes and Drug Encapsulation

The fabrication of TFs was performed using a thin-layer hydration sonication (TLHS) method [53,54,55]. In summary, a predetermined quantity of L-α-phosphotidycholine and Tween-80 (1%) was dissolved in 2 mL of volatile organic solvents, (chloroform:methanol 2:1) in a round bottomed flask. A rotary evaporator was used to evaporate the organic solvent under reduced pressure, at 45 °C, which is above the lipid transition temperature (24–37 °C) [56,57] to form a thin film of lipid. The thin film was subsequently hydrated with PBS (pH 7.4) for 2 h at room temperature (approximately 20 °C) at 150 rpm. The resulting multilamellar vesicles were then sonicated using a probe sonicator for 10 min to form the unilamellar transfersomes.

To encapsulate aspirin, the drug was mixed with l-α-phosphotidycholine and Tween-80 and dissolved in the same organic solvent (chloroform/methanol (2:1)) in the first step of the fabrication process. The aspirin to l-α-phosphotidycholine weight ratio in the formulation was 2:5. The encapsulation of drug models, including Au-NPs and the DiI labelling dye, was conducted using the same procedure employed for encapsulating aspirin.

##### Characterisation of Transfersomes

#### 2.2.2. Quantification of Aspirin-Encapsulated Transfersomes (TF–Asp)

In this study, one optimised formulation of TF–Asp was used. To accurately determine the encapsulation efficiency of aspirin, the unencapsulated aspirin was separated from the encapsulated aspirin through ultra-speed centrifugation. The supernatants were collected to measure the unencapsulated aspirin using HPLC (Agilent 1100 series, Santa Clara, CA, USA). A reversed-phase C18 column (Phenomenex Inc., Macclesfield, UK) (5 μm, 4.6 × 150 mm) was used as the stationary phase for an isocratic elution in a water–methanol–acetic acid (7:3:0.1) mobile phase, with a constant flow of 1 mL/min and a detection wavelength of 231 nm. The slope of the standard calibration curve for aspirin can be used to calculate the concentration of unencapsulated aspirin. The encapsulation efficiency (EE) of encapsulated aspirin was derived using the formula below:EE % = [(*Dt − Db)/Dt*] × 100
where *Db* is the measured amount of unencapsulated aspirin and *Dt* is the total amount of aspirin used in the initial step of transfersome fabrication.

#### 2.2.3. Size, Polydispersity Index and Zeta Potential Characterisation

Dynamic light scattering (DLS) was used to measure the size and polydispersity index (PDI) of TF. The physical stability of transfersomes was evaluated by monitoring the change in the size of TFs stored at three different temperatures (4 °C, 37 °C, and room temperature) over a period of 30 days. The zeta potentials of TFs were measured by electrophoretic light scattering (ELS). In this case, the TF and TF–Asp samples were resuspended in deionised water to avoid sodium ions in PBS buffer. The TF and TF–Asp samples were diluted 10 times and sonicated before testing to ensure that they were thoroughly dispersed.

#### 2.2.4. Fourier Transform Infrared Spectroscopy (FTIR) of Aspirin and Transfersomes

The TF–Asp samples were analysed using Fourier transform infrared spectroscopy (FTIR) to verify the encapsulation of aspirin inside the TFs.

The FTIR measurements were performed using the Perkin Elmer Spectrum Two UATR instrument. Samples were placed on the attenuated total reflectance (ATR) crystal and the spectra were recorded from 500 to 4000 cm^−1^ wave numbers. Spectra were recorded for aspirin, L-α-phosphotidycholine, and TF with encapsulated aspirin.

#### 2.2.5. Energy-Dispersive X-ray Spectroscopy (EDX)

Scanning electron microscopy (SEM) was used to study the morphology of the vesicles. EDX was then performed (5 kV) on areas identified from the SEM images across the TFs and Au-NPs encapsulated TF sample surfaces to chemically map the sample.

#### 2.2.6. TEM Investigation of TF (TF and TF-Encapsulated Au-NPs)

In this study, Au-NPs were employed as a representative drug model for encapsulation within TFs. The distinctive high contrast of Au-NPs against the background was utilised to clearly demonstrate their successful encapsulation within TFs. The morphologies of TF and TF-encapsulated Au-NPs were examined following negative staining using 2% uranyl acetate for 10 min. Transmission electron microscopy (TEM) (Philips CM12 TEM (FEI U. K. Ltd., Altrincham, UK) was used to investigate the morphology of the vesicles at 80 kV and images were captured with a Megaview III camera and AnalySIS software, version 3.2 (Soft Imaging System GmbH, Münster, Germany). TEM was performed at Central Biotechnology Services, Cardiff University.

##### Dissolution Studies of Encapsulated Aspirin

#### 2.2.7. In Vitro Release of Aspirin Using Dialysis Sacks

The in vitro release of aspirin from TFs was investigated using dialysis sacks (molecular weight cut-off of 12,000) to separate the released aspirin and the TFs which were impermeable to the TF. The rate of release of the encapsulated aspirin from TFs was compared with the transport of free aspirin (Free-Asp) through dialysis sack membranes into a phosphate-buffered saline (PBS) release medium.

#### 2.2.8. Stability of TF–Asp, A Time-Dependant Release Study

Drug leakage from TFs was tested via the time-dependent release of aspirin from TF stored at three different temperatures (4 °C, 25 °C and 37 °C degrees) for a period of 0, 15, 30, 60 and 90 days. The concentration of the TF–Asp with regard to the calculated encapsulation efficiency (67.5%) was 6.2 mg/mL. At each pre-determined time point (0, 15, 30, 60 and 90 days), 1 mL of each stored sample was transferred to a dialysis sack and the sacks were placed into beakers filled with 10 mL PBS as a release medium. The release of aspirin was monitored at different time points (0, 5, 15, 30, 45 and 60 min up to 10 h) and quantified using HPLC.

#### 2.2.9. Microneedle Fabrication

Low-density arrays of solid silicon microneedles (SMNs) were fabricated by Swansea University’s College of Engineering [37]. A combination of lithographic and deep reactive ion etch (DRIE) processing was used in the solid SMN fabrication process, eliminating the hollow bore etch step. An advanced three-step etching process was used to create the bevelled needle tip and the vertical microneedle shaft. Iterative BOSCH etching cycles were used to perform the deep silicon etch (using an SPTS Technologies Ltd, Newport, UK. DSi-V deep reactive-ion etching (DRIE) system) [38]. The etch rate used was approximately 10.75 µm/min.

The low-density microneedle array mask was designed with a microneedle pitch of 1000 µm. The SMN had an area of 7 mm by 1 mm and contained 4 microneedles.

Polycarbonate microneedles (PMNs) were fabricated via injection moulding using a micromachined metal mould. Solid polycarbonate microneedles were supplied by the College of Engineering at Swansea University as arrays of 16 microneedles (1000 μm insertion length and pitch size of 300 μm) with an area of approximately 1 cm by 1 cm.

##### Permeation Studies

#### 2.2.10. Skin Sample Preparation

Excised porcine skin samples excised from a white 8-week-old piglet was acquired from WetLab-Medmeat (Warwickshire, UK) in compliance with ABPR REG EEC 142/2011. The non-scalded porcine skin had an intact stratum corneum. The animals were delivered frozen, defrosted and with their hair shaved using clippers, and the skin was trimmed to a thickness of approximately 500 μm using a dermatome (Integra Life Sciences™, Padgett Instruments, Plainsboro, NJ, USA).

The side skin regions of the animal were selected to avoid the mammary areas of the belly side and ensure access to the largest and most consistently available skin samples. The skin samples were then stored in a freezer at −20 °C until the time of experiment. No skin samples with wounds, warts, or hematomas were utilised. Samples were used within two months of slaughter.

Human skin (breast skin from a 75-year-old female donor) was provided under local research ethics committee reference 08/WSE03/55).

Full-thickness human skin was used for the optical coherent tomography (OCT) analysis of MN skin penetration.

For the permeation studies, all formulations (TF, TF–Au and TF–Asp, with and without microneedles) were applied in the form of suspension, consisting of TF in PBS (phosphate-buffered saline).

#### 2.2.11. Optical Coherence Tomography (OCT)

The efficacy of the penetration of microneedles into skin layers was assessed using the OCT of human skin in vitro. OCT, providing qualitative data, guided our preference for human skin in testing due to its hairless and consistently and even surface compared to porcine skin. This choice minimises potential background interference, as porcine skin, with its uneven surface and hair presence, may introduce added complexities during OCT testing. SMNs and PMNs were inserted into the full-thickness (1000 µm thick) skin.

OCT experiments were undertaken at Cardiff University (at the School of Optometry and Vision Science). Human skin was cut into small square pieces (4 cm × 4 cm) and peripherally pinned to an underlying corkboard. The tissues were then treated with both types of microneedles (SMNs and PMNs) for 20 s each, at thumb insertion pressure (approximately 40 N per cm^2^). Subsequently, the microneedles were removed, and the treated area was directly placed under a light source with scans set to 500 frames over the skin area. OCT was also used to take in situ images of PMNs (in situ imaging was possible due to the transparency of the PMNs). Untreated skin was used as the control.

The scan frames were then collected and analysed using ImageJ for the validation of microneedle penetration through the stratum corneum and depth profile analysis.

#### 2.2.12. Application of TF-Encapsulated Au-NPs (TF–Au) for Qualitative Monitoring of Skin Permeation of TF

TF–Au were applied to porcine skin samples with and without the combination of SMNs and PMNs for the qualitative monitoring of the skin permeation of TFs in the skin layers using microscopy. In this study, the “poke and application (also referred to as a poke and patch)” method [35] was used, first applying the microneedles and then subsequently TF–Au. In this case, TF–Au was applied following skin application and the removal of the PMNs and SMNs to full-thickness skin samples. TF–Au were also applied without the assistance of the microneedles as a control. After treatment using TF–Au and microneedles, the skin samples were cut and transferred into test tubes. Treated skin samples were then fixed in 4% formaldehyde + 0.2% glutaraldehyde in 100 mM phosphate-buffered saline pH 7.4 for 24 h at room temperature and stored in PBS at 4 °C. Samples were thoroughly washed with reverse osmosis-filtered water (RO water) and post-fixed for 2 h with 0.1% tannic acid in 50 mM Tris-HCl buffer pH 8.0. The samples were then thoroughly washed in RO water and further fixed for 2 h in 10 mM uranium acetate. Uranium acetate was selected as the preferred metal complexing agent as it facilitates the subsequent silver development for localising Au-NPs. Gold nanoparticles were visualised at the light microscopic level by silver development. Sections were treated with Newman and Jasani’s physical developer [58] for 25 min.

#### 2.2.13. Franz Cells In Vitro Permeation Studies—Using Transfersomes and Microneedles

Vertical Franz diffusion cells (Soham Scientific, Ely, UK) with an active area of 0.64 cm^2^ were used to assess the in vitro permeation. The receptor chambers were filled with degassed PBS (pH 7.4) (selected to mimic the pH of the human body) and the magnetic stirrer was set at 400 rpm to maintain the receptor solution homogeneity. The temperature of the Franz cells was set and maintained at 32 °C (skin surface temperature [59]). A poke and application method was used to treat the skin and apply the TF formulation. Porcine skin samples were treated using the microneedles applied with a 40 N/cm^2^ pressure (approximate to finger pressure), consistent for both SMNs and PMNs. However, it should be noted that SMNs are able to penetrate skin at far lower forces (as low as 2 N/cm^2^). SMNs were applied four times iteratively to achieve the same number of pores as PMNs (16 pores). Following the removal of the microneedles, the porcine skin samples were sandwiched between the donor and receptor Franz cell chambers. The dermal side was in contact with the receptor medium.

The original quantity of aspirin used to load into the formulation was 80,000 µg in 200,000 µg of phosphatidylcholine, and the resultant vesicles were subsequently suspended in 4000 µL of the buffer. The average encapsulation efficiency was 67.5%, thus equating to a total drug load of 54,000 µg. From this 4000 µL buffer, 500 µL of the TF–Asp was introduced to the donor compartment of the Franz cell. This equates to 6750 µg aspirin in 500 µL.

Samples including an unencapsulated free-aspirin solution (Free-Asp) and TF–Asp were tested, with and without the assistance of SMNs and PMNs. The concentrations of Free-Asp and TF–Asp were 750 µg/500 µL (or 1172 µg/cm^2^) and 6750 µg/500 µL (or 10,547 µg/cm^2^), respectively. The lower concentration of Free-Asp is related to the limited aqueous solubility of aspirin (1500 µg/mL). Nine Franz cell replicates were used for each sample. Samples were collected at time points of 5, 15, 30, 60, 120, 240 and 1440 min. Sample volumes were replaced with fresh PBS solution. HPLC was used to quantify the concentration of permeated aspirin in the receptor compartment of the Franz cell at each time point. The cumulative permeated amount for each drug sample was plotted against time, and permeation percentage was calculated.

#### 2.2.14. Cytotoxicity Tests of TFs, with and without Aspirin

The viability of human fibroblasts was assessed in the presence of different concentrations of Free-Asp, TF–Asp and TF (Table 1 and Table 2) compared to the viability of untreated control cells. The range of aspirin concentrations (Table 1) was selected to assess cytotoxicity and designed to be within the effective therapeutic range [60].

Different concentrations of Free-Asp and the identical concentration of aspirin, encapsulated inside TFs (Table 1), were added to different sets of wells. Control TFs with phospholipid concentrations which would be required to encapsulate different concentration of aspirin, respectively (Table 2), were also added to a different set of wells in order to test the cell viability in the presence of control blank particles. Cell growth was quantified at 24 h, 48 h and 96 h of treatment using AlamarBlue cell viability assays.

The proliferation of the cells exposed to each group of samples with different concentrations of Free-Asp, TF–Asp, and blank TFs were compared.

#### 2.2.15. Intracellular Uptake of TFs by Human Fibroblast Skin Cells

TFs were labelled with DiI Dye (TF–DiI) to create contrast between the TFs and the background (cells) (see Section 2.2.1). A total of 10,000 cells of each cell line of human skin fibroblasts were seeded in each well of the 96 well plates and incubated at 37 °C overnight. Non-toxic concentrations of TFs and TF–DiI (0.2 mg TF/mL) were optimised and added to each well. The plates were then incubated at 37 °C inside the incubator. Subsequently, one microplate at each time point (0, 2, 4, 6, 8, 10, 24 h) was removed from the incubator for analysis using confocal microscopy.

#### 2.2.16. Statistical Analysis

The cumulative Franz cell permeation data and permeation percentage were expressed as a mean ± standard error (SE) based on 9 Franz cell replicates. A one-way ANOVA was conducted to compare the statistical difference between the results, using the software Prism (version 6). All other results in this study used a mean ± standard deviation (SD) based on 3 replicates. For all the statistical analyses, means were indicated to be statistically different when *p* < 0.05.

## 3. Results

### 3.1. Characterisation TF

#### 3.1.1. SEM Characterisation of TFs

The SEM analysis of freeze-dried TFs was used to characterise the surface morphology, shape, and diameter of the TFs. The size distribution of the multilamellar vesicles (MLVs) ranged from 200 nm to 500 nm in diameter (Figure 1a). These MLVs are known to have multiple lipid bilayers surrounding the aqueous core. To achieve the desired size of the vesicles (less than 200 nm), the MLVs were further processed through sonication. Post sonication, the resulting spheroid TFs (Figure 1b) had greater uniformity in terms of size, shape and morphology. Obtaining sub-100 nm sizes for the TF particles was critical in producing TFs that were small enough to permeate through skin pores as small as 20 nm. TFs are ultra-flexible, and thus, under 80–100 nm [17,61]. TFs are suitable for transdermal drug delivery.

#### 3.1.2. Size Analysis

The TF size was confirmed by dynamic light scattering (DLS) (Table 3) to be less than 100 nm. Considering some limitations of the thin-film hydration sonication method in terms of the monodispersity of the particles, the average polydispersity index of TFs remained close to 0.2, indicating a reasonably uniform size distribution of the vesicles for drug encapsulation (Table 3). In drug delivery applications, a PDI of 0.3 or below is considered to be an acceptable uniformity of vesicle size distribution [62]. The size stability of TFs stored at three different temperatures (4 °C, 25 °C and 37 °C) after 30 days showed that TFs are quite stable at fridge temperature (4 °C). However, storage temperatures of 25 °C and above significantly affect the size, PDI and disruptions of TFs (Table 3). This is due to the development of membrane fluidity, as well as the expansion and destabilisation of the vesicles’ structure [63,64].

The encapsulation of aspirin inside TFs did not significantly affect the size of the TFs (Table 4).

In this study, phosphatidylcholine with 99% purity was used to fabricate the TFs and TF–Asp. Phosphatidylcholine is a common phospholipid found in biological membranes and is known to have a zwitterionic nature related to the positively charged choline and negatively charged phosphate groups. This results in a zeta potential that is close to neutral (Table 4). The zeta potentials of TFs indicate the low resistance to aggregation of the vesicles and the instability of the TFs due to the low electrostatic repulsion between the particles in the sample. Zeta potentials indicated less aggregation with aspirin encapsulation (Table 4). Aspirin is a weak acid and greater acidity leads to vesicles with a more positive zeta potential [65].

#### 3.1.3. FTIR Characterisation of TF–Asp

The Fourier transform infrared spectroscopy (FTIR) inspection of TF–Asp confirmed the presence of aspirin in the samples (Figure 2). The aspirin peaks were clearly present in the encapsulated aspirin sample, at 1754 cm^−1^, 1684 cm^−1^, 1606 cm^−1^, 1459 cm^−1^, 1374 cm^−1^, 1308 cm^−1^, 1222 cm^−1^, 1186 cm^−1^, 1000 cm^−1^ and 917 cm^−1^. Peaks are present in the TF-encapsulated aspirin (TF-Asp) samples—and not present in the phospholipid-only sample—at 1607 cm^−1^, 1374 cm^−1^, 1308 cm^−1^, 1222 cm^−1^ and 917 cm^−1^. Aspirin peaks are due to the stretching and bending vibrational absorption modes of aspirin—predominantly related to the vibrational absorption frequencies of the carbonyl (carboxylic acid and ester groups) and aromatic ring structures in aspirin. Aspirin also has a broad IR absorption between 2500 and 3000 cm^−1^ due to the OH group. However, this broad peak cannot clearly be discerned in the spectrum of the encapsulated aspirin.

#### 3.1.4. Drug-Load, Encapsulation Efficiency, Drug Release and Stability Study of TF–Asp

The initial drug-loading was 40% at a TF:aspirin ratio of 2:5 *w*/*w* (80 mg aspirin to 200 mg phospholipid). The encapsulation efficiency of aspirin inside TFs was found to be 67 ± 2%, suggesting a drug-loading of 26.8%

Figure 3a shows that TFs exhibit the controlled release of Free-Asp over a 600 min period, where most of the Free-Asp (4 mg out of 5 mg total aspirin) from the dialysis sack was released. As the Free-Asp release burst occurs during the first 120 min and becomes essentially saturated after 600 min, the experiment was discontinued at this time point. The slower and controlled release of aspirin out of TFs is shown over the initial 600 min period (Figure 3a) and during an extended period of 10 days (Figure 3b). If the experiment had been continued, further aspirin would have been released at a slow and consistent rate. The burst release of aspirin from the TFs was 14% (0.7 mg out of 5 mg) after 120 min.

The stability studies of TF–Asp show no significant difference between the release of aspirin from the baseline (fresh TF–Asp) (average of 1.43 ± 0.05 mg release) and the release from TFs stored for 15 (1.6 ± 0.2 mg), 30 (1.80 ± 0.02 mg) and 60 (2.1 ± 0.06 mg) days, which confirms the stability of TFs at 4 °C temperature for 60 days. There was, however, a significant difference from the baseline after 90 days storage of TF–Asp at 4 °C (*p* value < 0.5). Even though the release of aspirin from TFs held at 4 °C for 90 days was analysed to be statistically significant (2.6 ± 0.05 mg), the actual quantitative aspirin release at this temperature was minor (Figure 4a). The percentage of aspirin remaining inside the TFs after each time period shows that there was negligible leakage of aspirin from the TFs after 90 days of storage at a temperature of 4 °C. The aspirin contents inside the baseline batch of TFs were 77 ± 0.7%, which decreased to 75 ± 2%, 72 ± 0.8%, 67% and 59 ± 2% after 15, 30, 60 and 90 days of storage, respectively (Figure 4a).

Stability tests on TF–Asp stored at room temperature (25 °C) showed that this higher temperature had a significant effect on the release of aspirin from the TFs. The release of aspirin from the TFs increased from 1.43 ± 0.05 mg (baseline) to 1.8 ± 0.03 mg, 2.44 ± 0.34 mg, 3 ± 0.09 mg, and 3.6 ± 0.07 mg after 15 days, 30 days, 60 days and 90 days of storage, respectively. The aspirin content inside the TFs stored at this temperature (25 °C) decreased from 77 ± 0.7% to 71 ± 0.4%, 60 ± 5%, 54 ± 1.4% and 43 ± 1% after 15 days, 30 days, 60 days, and 90 days of storage, respectively (Figure 4b). This indicates the reduced stability of TFs stored at this temperature in terms of aspirin release.

Further increasing the temperature to 37 °C caused the further release of aspirin from the TFs from baseline at 1.43 ± 0.7 mg to 2.6 ± 0.2 mg, 3.3 ± 0.2 mg, 3.7 ± 0.2 mg and 4 ± 0.02 mg from TFs stored for 15, 30, 60, and 90 days, respectively. The percentage of aspirin content inside the TFs decreased from 77 ± 0.7% to 59 ± 3.2%, 48 ± 3%, 42 ± 2% and 37 ± 2% after being stored for 15, 30, 60 and 90 days, respectively. This confirms that, at 37 °C, the TFs are not stable in terms of aspirin release (Figure 4c) and storage at 4 °C is recommended.

#### 3.1.5. Encapsulation of Au-NPs inside TFs for Characterisation

To assess the transportation of TFs within skin layers, Au-NPs were encapsulated inside TFs, and skin samples treated with TF–Au were characterised using TEM. Figure 5a shows a cluster of non-encapsulated Au-NPs (of sizes in the range of 5–20 nm). The TEM images (Figure 5b,c) show spherical TFs with sizes varying from 100 to 200 nm, which is consistent with previous SEM and DLS (Figure 1 and Table 1 and Table 2). Figure 5b shows images of the two merged blank TFs, and it can be clearly seen that there are no Au-NPs encapsulated inside the TF, whilst it can be observed in the clusters of encapsulated Au-NPs that the TFs encapsulate Au-NPs (in Figure 5c). Clear TEM evidence of TF–Au strongly suggests that the encapsulation of drugs and molecules inside TFs is highly feasible.

The EDX elemental mapping of an individual TF–Au confirmed the presence of Au in the TF–Au samples. Figure 6 shows scans for control TFs with no encapsulation of Au-NPs, where no elemental gold signal was detected. Elements including carbon, oxygen, phosphorous and sodium—which all emanate from the phospholipid membrane—as well as aluminium—which is from the foil substrate are detected. In the sample containing AuNPs, a gold signal was also detected—which is representative of the Au-NPs (Figure 7)—were detected.

Figure 8 shows the point EDX scans moving from outside the TFs (aluminium substrate) into the centre of the TF region, where the Au signal increases from 0 to 10.07 atomic percentages, respectively, as it moves towards the centre of the TF.

### 3.2. Microneedle Characterisation (Silicon and Polycarbonate)

#### 3.2.1. Morphology of Microneedles

High-aspect-ratio SMNs of 700 μm in length and with a pitch size of 1000 µm were fabricated with near-vertical sidewalls (re-entrance angle of 94° at length of 700 µm) (Figure 9a). The pitch aspect ratio (ratio of the needle pitch to the needle radius) of low-density microneedles with microneedle shaft diameters of 240 µm was calculated to be 8. It has been suggested that a pitch ratio greater than 2 is required for pain-free microneedle skin penetration. Thus, the low-density microneedles should penetrate adequately.

The robust PMNs are pyramids in shape with an average length of 1000 µm and a pitch size of 300 µm, with a uniform size in each array of 16 PMNs Figure 9b).

#### 3.2.2. Penetration Efficacy of Microneedles Using Optical Coherence Tomography (OCT)

The disruption of the stratum corneum and the penetration into dermal layer by SMNs can be clearly seen in Figure 10b compared with untreated human skin (Figure 10a). The penetration of the sharp SMNs is highly perpendicular to the skin layers and creates a narrow vertical insertion profile with minimal indentation and skin damage. In contrast, PMNs penetrate into the epidermal layer with a higher degree of indentation and skin deformation (Figure 10c).

#### 3.2.3. Characterisation of Permeation of TF–Au into Skin Layers

The permeation efficiency of TF–Au into skin layers with and without the aid of microneedles was visualised using silver development technique.

The porcine skin treated with TF–Au shows the majority of the TF–Au permeated into the stratum corneum and top epidermal layer, and to a lesser degree, some permeation into the deeper epidermal layer of the skin sample (Figure 11a). Both types of microneedles (SMNs and PMNs) penetrated into the deeper layers of the skin (epidermal and dermal layers) and significantly enhanced the delivery of the TF–Au relative to the samples of TF–Au applied without using microneedles (Figure 11b,c and Figure 11d,e, respectively). The TF–Au appears to be diffused into the surrounding tissue layers after penetration using the microneedles. The average penetration depth of TF–Au inside the skin layers was approximately 100–150 μm.

### 3.3. In Vitro Permeation Studies

Comparisons between Free-Asp and Free-Asp combining SMNs and PMNs used the same concentration of aspirin (750 µg/500 µL), whilst comparisons of TF–Asp and TF–Asp combining SMNs and PMNs used the same concentrations of aspirin and TFs (6750 µg/500 µL). The higher concentration of aspirin made possible using TF encapsulation allows for significantly more aspirin to be applied to the skin surface. Figure 12a shows no significant difference in the permeability of Free-Asp and TF–Asp after 1 h. However, there is a significant difference found between Free-Asp and TF–Asp after 3 h, 5 h and 24 h, which confirm the enhancement in the permeability of aspirin using TFs. Figure 12b compares the permeability of microneedle-assisted TF–Asp and Free-Asp without microneedle-assisted delivery in Figure 12a. Overall, the microneedles significantly increased the permeability of TF–Asp and Free-Asp. The average permeability of microneedle-assisted TF–Asp was found to be higher (approximately four times) than the permeability of microneedle-assisted Free-Asp. The cumulative permeation of the TF–Asp samples was 2706 ± 548 μg using SMNs and 2259 ± 297 μg using PMNs, which illustrates that there is no significant difference between the use of SMNs and PMNs in terms of the enhancement of aspirin delivery. The in vitro permeation of Free-Asp samples was found to be 696 ± 85 μm using SMNs and 690 ± 86 μm using PMNs. This confirms that the combination of microneedles (SMNs and PMNs) with TF–Asp significantly enhances the permeation of aspirin across the skin.

Figure 13a shows the percentage permeation of TF–Asp versus Free-Asp across porcine skin. There is a significant difference between TF–Asp and Free-Asp after 5 h and 24 h. The permeation percentage is lower in the TF–Asp samples (3 ± 0.2%) than the Free-Asp (7 ± 5%) after 24 h. However, the quantity of permeated aspirin in the TF–Asp samples was significantly higher (206 ± 17 μg) than that in the Free-Asp sample (57 ± 11 μg) after 24 h. This confirms that the TF–Asp allows a significantly larger mass of aspirin (around four times) to penetrate into the skin layer relative to Free-Asp.

The percentage permeation of encapsulated aspirin using SMNs and PMNs was 40 ± 8% and 33 ± %4, respectively (Figure 13b). The percentage permeation of Free-Asp+SMN and Free-Asp+PMN was 93% ± 11 and 92% ± 10. Despite the percentage of permeated aspirin being higher for the Free-Asp sample compared to the TF–Asp, the actual mass of aspirin transported across the skin barrier is significantly higher (approximately four times) for encapsulated aspirin relative to free aspirin. This indicates that the microneedle-assisted transdermal delivery of TF–Asp is a highly effective method for administering an increased aspirin dose into skin compared to Free-Asp.

### 3.4. Viability of Human Skin Fibroblasts Exposed to Different Concentrations of TFs, Free-Asp and TF–Asp

AlamarBlue assays were used to quantify the viability of human skin fibroblasts exposed to Free-Asp and TF–Asp (aspirin doses of 400 μg/mL, 200 μg/mL, 100 μg/mL, 50 μg/mL and 25 μg/mL) (Figure 14a,b) and control TFs with the same phospholipid concentrations required to encapsulate the concentration of aspirin (0.8 mg/mL, 0.4 mg/mL, 0.2 mg/mL, 0.1 mg/mL and 0.05 mg/mL) (Figure 14c).

The cytotoxicity effect of the Free-Asp and the TFs was found to be dose-dependent with respect to human fibroblasts cells (i.e., as the concentration of aspirin and the amount of TFs increased in the sample, the cytotoxicity on human fibroblasts increased). Low doses of aspirin (25 and 50 µg/mL) showed no cytotoxic effect on the fibroblasts. It appeared that the aspirin cytotoxicity is initiated at concentrations of 100 µg/mL and higher (Figure 14a,b). Low concentrations of TFs (0.1 mg/mL and 0.05 mg/mL) also showed no cytotoxic effect on human fibroblasts. However, the TF concentration at 0.2 mg/mL and above showed some cytotoxicity on the fibroblasts (Figure 14c).

### 3.5. Intracellular Uptake of TFs by Human Skin Fibroblasts

Factors influencing the uptake of particles include concentration, particle size and time [66,67]. Previous studies have shown that uptake into non-phagocytic cells (human fibroblasts) strongly depends on the TF size with an optimum uptake of vesicles with an average diameter of 50–100 nm [66,68]. The uptake of TF–DiI (average size of 100 nm and a TF–DiI concentration of 0.05 mg/mL) after 24 h incubation was observed using a confocal microscope (Figure 15) (red intensity represents the TF–DiI and the blue intensity represents the nuclei of the cells). From the figures, it seems that the TFs were localised to a greater extent inside the cytoplasm of the cells.

The uptake of TF–DiI by human fibroblasts cells was observed over time periods of 2 h, 4 h, 6 h, 8 h, 10 h and 24 h using a confocal microscope (Figure 16). The uptake of TFs by human fibroblasts appeared 2 h after the treatment (low red intensity) and significantly increased after 24 h (high red intensity).

## 4. Discussion

TFs have been fabricated and characterised and applied in combination with PMNs and SMNs for transdermal drug delivery applications. The thin-film hydration sonication method is a well-known method for vesicle production at the lab scale, and produces reasonable uniformity in terms of particle size. However, the method has some limitations (residual solvents and limited monodispersity) affecting the size control of the vesicles [69], making this method less suitable for scaling up. The size and stability of the DLS measurements of TFs showed the increased aggregation and coalescence of TFs at higher temperatures and that TFs are stable at 4 °C for storage up to 90 days, which was tested. TEM analysis showed that Au-NPs could be successfully encapsulated inside TFs, as validated using EDX. FTIR was used to validate aspirin encapsulation inside TFs. The optimised TF particles have an encapsulation efficiency of 67.5% for encapsulating aspirin.

The zeta potential for TFs fabricated using 99% pure phosphatidylcholine is essentially neutral but can be easily affected by different factors including impurities and pH. As a weak acid, aspirin encapsulation modifies the TF charge to yield positively charged vesicles.

The controlled release of aspirin over a period of 10 days was observed. An initial burst of release within 2 h is followed by a constant release rate over time. Factors affecting drug release from TFs include the presence and percentage of surfactants, the composition of TFs, physical/chemical drug properties and the percentage of encapsulated drug [17,22,28]. An increase in the concentration of surfactant destabilised the TFs and enhanced the drug release [52]. A more acidic environment produced a burst release of aspirin from the TFs, as cationic ions interacting with anionic heads and the group of phospholipids destabilise the vesicle membrane by the protonation of the functional head group.

This induces morphological changes in the vesicle bilayer and consequently causes the burst of aspirin release. Drug release from TFs after storage at 4 °C was relatively low, with the majority of the drug remaining within the TFs over a period of 90 days. After storage at higher temperatures (25 °C and 37 °C), the ratio of drug released to drug remaining was increased due to the lower stability of the TFs after being stored at these temperatures. The instability of TFs stored at higher temperature is related to the lipid transition temperature and the deformation of the TF structure at high temperatures [26,64,70].

The three possible mechanisms for the release of aspirin from TFs are (i) release from the intact TFs that have been transported into the skin layers; (ii) aspirin released onto the skin surface by the disruption of the TF lipid bilayer membrane and the release of the TF contents onto the skin surface; and (iii) a combination of these aforementioned mechanisms. Studies of microneedle-assisted TFs encapsulating Au-NPs showed the enhancement of vesicle transport into deeper skin (dermal layer) through microchannels created by microneedles and with the diffusion of TF-NPs into tissue adjacent to the microneedle injection site [71,72,73].

Transdermal drug permeability is affected by the physicochemical properties of drug formulations such as molecular size partition coefficient, melting point, solubility and ionisation [74,75,76]. Amongst all these factors, the partition coefficient of a compound is critical in understanding the most probable skin penetration pathway for molecules to take. It is hypothesised that hydrophilic molecules partitioned into the hydrated keratin-filled keratinocytes, while hydrophobic molecules favoured partitioning into the lipoidal bilayers. Consequently, the hydrophilic molecules exhibited a higher likelihood of penetrating through the intracellular pathway, while hydrophobic molecules displayed a greater tendency for permeation through the intercellular pathway [77].

Numerous studies suggested that an increase in the hydrophobicity of a molecule correlates with enhanced permeation into the stratum corneum [78,79]. This implies that the stratum corneum lipid bilayer serves as a rate-limiting barrier for the permeation of hydrophilic molecules through this skin layer. However, with further increases in the hydrophobicity of molecules, a new rate-limiting barrier emerges, affecting partitioning into the deeper layer of skin (dermis) characterised by a more aqueous environment [80,81].

The optimal condition for a drug to permeate across intact skin, as indicated by the most favourable partition coefficient, fell within the intermediate range of Log P (octanol/water) which is from 1 to 3 [82,83,84,85]. The encapsulation of drugs inside phospholipid bilayer vesicles, such as TFs, solves the problems associated with the hydrophilicity and hydrophobicity of drugs and enhances the permeability across the skin. Microneedles are very effective in creating channels for drug permeation into deeper skin layers. However, hydrophobic drugs still present difficulties with respect to permeation in the more hydrated dermal layer. Thus, the combination of TFs and microneedles addresses this issue.

Ionisation plays a significant role in influencing the skin permeability of drugs. Ionised drugs are prone to permeation via the shunt route, while a non-ionised drug is anticipated to be primarily transported through the lipid intercellular pathway [86]. Ionised molecules have the potential to attract to the polar head groups of the lipid domains in the stratum corneum and can dissociate to varying degrees depending on the pH of the skin and formulation. Consequently, in general, ionised molecules are poor transdermal permeants [86,87,88]. Aspirin is a weak acid, and thus, it tends to ionise in a high pH aqueous environment by losing an electron [89]. Ionised drugs do not pass through biological membranes [90,91,92]. Aspirin is predominantly non-ionised in a low pH environment, like the stomach (pH = 2), and thus, it permeates the biological cell membranes into the blood stream. Ionised drugs such as aspirin therefore have difficulty in passing across the skin. The encapsulation of aspirin resolves this issue as it allows aspirin to be used for the topical and transdermal application of the TF formulation. Aspirin is known as one of the most acidic drugs which causes quite severe side effects in the gastrointestinal tract. Microneedle-assisted TFs encapsulating aspirin is thus a highly desirable method for delivery, which resolves the difficulties of oral aspirin administration, providing safe delivery. Drug solubility constitutes another crucial factor that influences the skin permeability of drugs. Drugs characterised by poor aqueous solubility, as exemplified by aspirin, are not fully soluble in the aqueous formulation. Consequently, the quantity of the topically applied drug presented on the skin surface was relatively limited. Encapsulation is an ideal way to resolve this issue in the case of aspirin.

In excised skin samples, blood circulation and metabolic activity are absent, regeneration has stopped, and immune system has ceased [86]. Thus, molecular permeation is primarily based on the diffusion of the molecules into the Franz cell aqueous receiver. This study showed that TFs enhanced the delivery of aspirin across the skin by approximately four times. The significant enhancement in permeation was observed for both free and encapsulated drugs using microneedle-assisted delivery. This was the case for both SMNs and PMNs.

Microneedle-assisted delivery yielded approximately 13- and 10-fold increases in permeation of encapsulated aspirin using SMNs and PMNs, respectively. This clearly shows the advantages of a combined microneedle and TF delivery system, where microneedle-assisted delivery produced an even more marked increase in permeation compared to using the TFs alone. This is related to the larger skin pores opened by the microneedles, which facilitate the transport of larger particles (TFs) through the skin barrier. In turn, the augmented transport of TF allows an increased concentration of drug to be delivered into skin. The difference between the two types of microneedles is primarily the shape of the indentation profile. For SMNs, there is a narrow shaft with the bevel shape of the needle transferred into the skin layer. PMNs show a pyramidal indentation with clear skin disruption only at the tip of the microneedle. However, this still appears to be sufficient for effective transport into skin layers.

The permeation study was used to demonstrate that the aspirin dose that was effectively transported across the skin could be within a therapeutic dose range of 3–10 µg/mL [60]. The optimum system, using the silicon microneedles combined with TF–Asp, could deliver 23% of the 3 µm/mL minimum therapeutic aspirin dose using an application over the 0.64 cm^2^ active area of the Franz cell. Thus, its application on a larger active area of 2.74 cm^2^ could achieve a minimum required delivery dose of 3 µm/mL. However, this limited study does not take into account in vivo conditions, which would undoubtedly affect the delivery dose.

Currently, the limitation of the combined TF-MN approach is related to the requirement for two separate steps (poke and application). In the future, this method could be improved by combining these steps in a single process.

Microneedles have been reported as a relatively safe therapeutic procedure for transdermal drug delivery systems. Whilst multiple use of derma rollers shows a risk of skin infection [93], we found no report of infection using single-use microneedles. Moreover, there are some reports of microneedles producing minor skin inflammation, erythema, pain, and skin irritation immediately after use. More severe adverse events are relatively rare [94].

The in vitro cytotoxicity studies showed that aspirin incurs cytotoxicity at concentrations of ≤100 μg/mL. In the in vitro study using the recommended dose for aspirin, a high degree of cytotoxicity was observed in the human skin fibroblasts cells. The same effect would not be expected for in vivo studies, as the drugs would be up-taken by blood vessels and lymph nodes; thus, the drugs would be cleared from the localised skin site, avoiding the build-up of cytotoxic levels. TFs did not reduce the cytotoxicity of aspirin, since a degree of cytotoxicity was exhibited by the TFs themselves at concentrations of ≤0.2 mg/mL.

The targets for many therapeutic molecules are localised in the subcellular compartment of the cells. Thus, the interaction between the drug carriers and the cells, and the cellular uptake of vesicles and particles are critical aspects of effective drug therapy. The cellular uptake of TFs was higher over time and confirmed that the majority of TFs localise in the cytoplasm of the cells. The localisation of vesicles in cells is critical, since the cytoplasmic localisation of particles would have less interference with more complex molecules in cells such as DNA, and thus cause less cytotoxicity. Although the different pathways of cellular uptake, such as endocytosis, fusion and lipid-exchange have been characterised, their role in the controlled delivery of pharmaceuticals is still not quantified. Various factors, such as the size, shape, surface charge and properties of the drug carriers affect the endocytosis of vesicular carriers [95,96]. Confocal microscopy, a widely used technique for imaging the cellular uptake of nanoparticles, was used to image the internalisation of fluorescently labelled TFs (TF–DiI) in the 2D molecular cell cultures of human skin fibroblasts. The study by Prabha S et al. on the internalisation of small- (average of 70 nm) and large-size (average of 200 nm) PLGA NPs encapsulating plasmid DNA (encoding luciferase protein) reported that the smaller size NPs increased transfection by 27 times more than the larger NPs in the HEK-293 (human embryonic kidney) cell line. This indicates the uptake of small PLGA NPs was higher than that for the large particles [97].

An overview of cellular uptake mechanisms by Salatin S. et al. stated that the more facile internalisation of smaller particles is due to a larger surface area relative to the same mass of larger particles. The high surface area allows the small particles to have greater contact with the biological membrane [98]. Zhang Y. et al. studied the effect of skin viability on the transdermal delivery of liposome-encapsulated Psoralen (lipo–Pso), reporting that a decrease in cell viability (in skin layers), increased the delivery of lipo–Pso across the skin. This study stated that the increase in lipo–Pso transport could be due to less cellular uptake and thus less complexity in the permeation route across the skin [99]. This study indicated that the skin viability effects should be taken into account in terms of the evaluation of the drug carrier permeability across the skin [99]. The microneedle-assisted delivery of TF provides a faster delivery route, transporting the TFs closer to the targeted area (circulation capillaries) and avoiding the large uptake of TFs by cells. However, if the aim of the application is to treat skin conditions (e.g., eczema and psoriasis), then the uptake of TFs by local cells is favourable and beneficial.

The painless or minimally invasive nature of microneedle applications can increase patient compliance, especially in populations with needle phobia or those requiring frequent injections (like diabetics). Improved compliance can lead to better treatment outcomes and patient satisfaction. The combination of TF–drug formulations and microneedle-assisted delivery is suggested to be of interest to the pharmaceutical industry, which views microneedle technology as a key area of growth, offering improved patient experience and compliance, and new market opportunities. This is evidenced by the large numbers of patent filings by major companies like Sorrento Therapeutics, Kimberly Clark, and Hisamitsu Pharmaceutical. Examples of microneedle systems include Sorrento Therapeutics’ melanoma treatment [100] and skin health treatment from Johnson & Johnson and Raphas [101]. Additionally, Zosano Pharma have developed a titanium microneedle patch for migraine relief [102]. Microneedle treatments combined with new formulations, although currently expensive, will reduce in cost with mass production and are likely to increase in terms of their number of applications. In addition, the economic analysis of microneedles combined with encapsulated formulations must take into account the holistic patient treatment regimen. This includes fewer complications and costly health interventions resulting from the increased compliance related to microneedle/TF treatments.

## 5. Conclusions

The encapsulation of drugs with low water solubility inside TFs resolves the solubility issues of hydrophobic drugs and provides optimal hydrophilicity/hydrophobicity for the permeation of drugs through skin layers. An aspirin encapsulation efficiency of 67.5% suggests a promising method for the encapsulation of drugs with similar physicochemical characteristics. TF encapsulation also addresses the issue concerning the application of weak acid drugs on skin by providing a more suitable surface charge for the skin permeation of drugs. TFs facilitated the controlled release of aspirin over time and provided a shelf life of 90 days when stored at 4 °C. The microneedle-assisted delivery of TF–Asp significantly enhanced the permeation of aspirin by 13- and 10-fold using silicon and polycarbonate microneedles, as opposed to 4-fold enhancement when using TF alone. Microneedle-assisted TFs represent a very effective transdermal drug delivery system for the delivery of different types of drugs and for different applications.

## Figures and Tables

**Figure 1 pharmaceutics-16-00057-f001:**
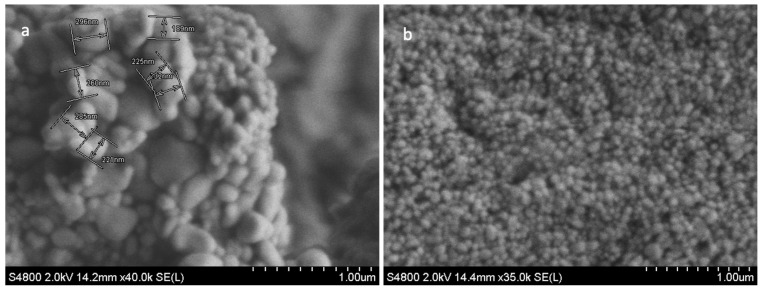
SEM images of MLVs (**a**); and blank TF (**b**). Image (**a**) shows the non-uniform morphology structure of MLV TFs, and the variation in the vesicles size and shapes ranging from 200 to 500 nm. The SEM image (**b**) shows less variation in the size and shape of TFs, and the size ranged from 80 to 100 nm (**b**). Scale bar = 1.00 µm.

**Figure 2 pharmaceutics-16-00057-f002:**
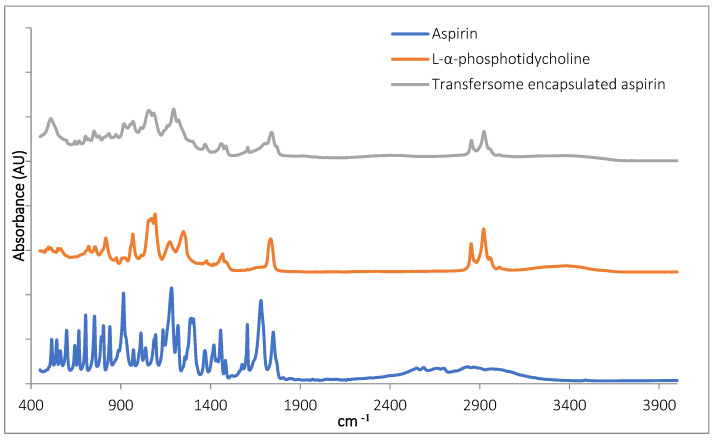
FTIR of aspirin (blue), L-α-phosphatidylcholine (red) and TFs with encapsulated aspirin (grey).

**Figure 3 pharmaceutics-16-00057-f003:**
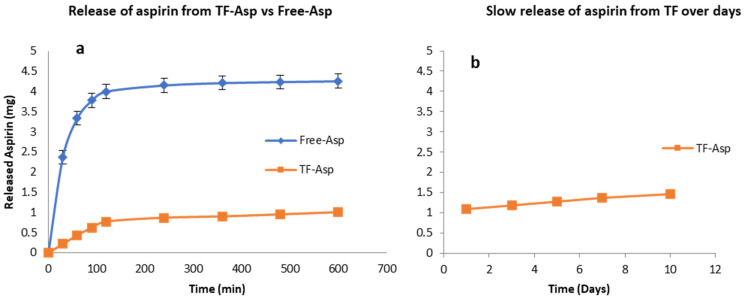
Release of aspirin from TFs (**a**,**b**) versus free aspirin transport (**a**). TFs control the release of aspirin. The release rate of encapsulated aspirin is slower than for the free drug. N = 3.

**Figure 4 pharmaceutics-16-00057-f004:**
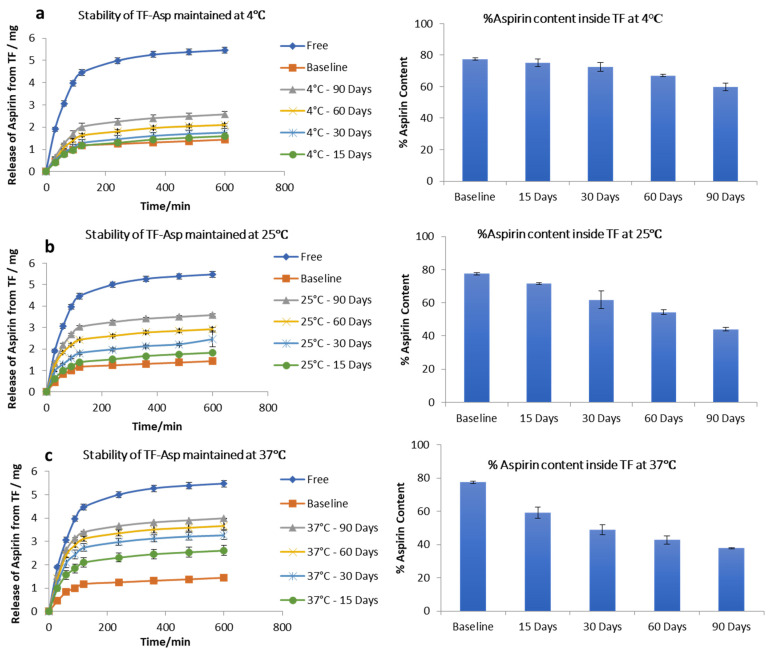
Stability of TFs stored at 4 °C (**a**), 25 °C (**b**) and 37 °C (**c**). The graphs show the release of aspirin from TFs and the %aspirin content inside the TFs at each time point compared to the baseline and free aspirin. N = 3.

**Figure 5 pharmaceutics-16-00057-f005:**
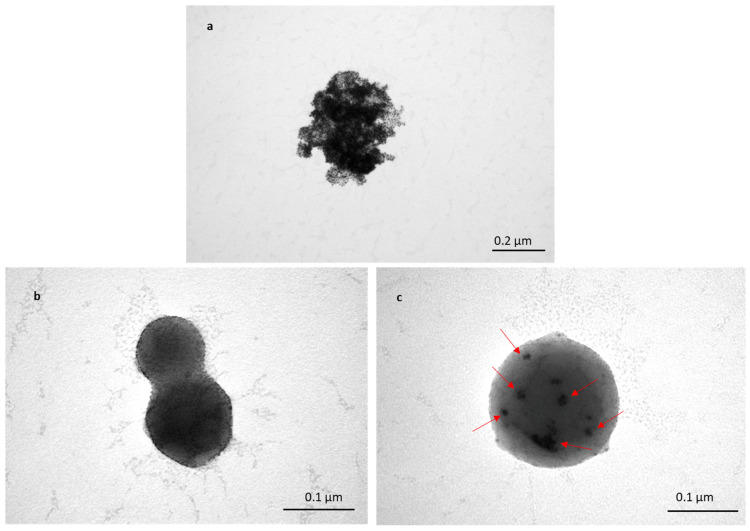
TEM image of Au-NPs with sizes of 5–20 nm (**a**); two merged blank TFs (**b**); and TF–Au (**c**). Arrows indicate the encapsulated Au-NPs inside the TFs.

**Figure 6 pharmaceutics-16-00057-f006:**
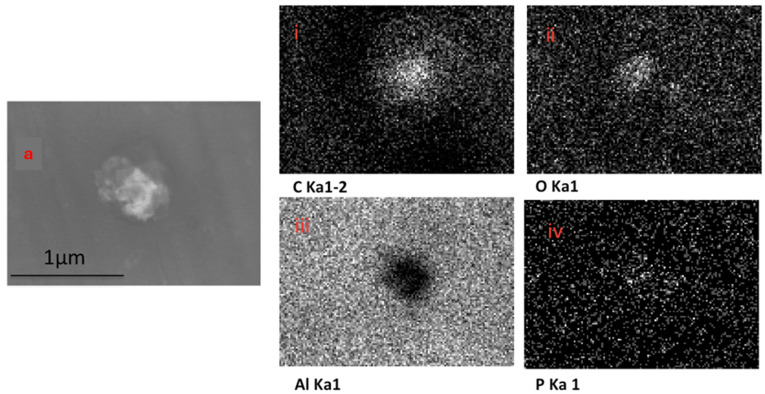
SEM backscatter of the disrupted blank TFs on aluminium foil (**a**) and the EDX mapping of this TF (**i**–**iv**), which shows no elemental signal for Au-NPs.

**Figure 7 pharmaceutics-16-00057-f007:**
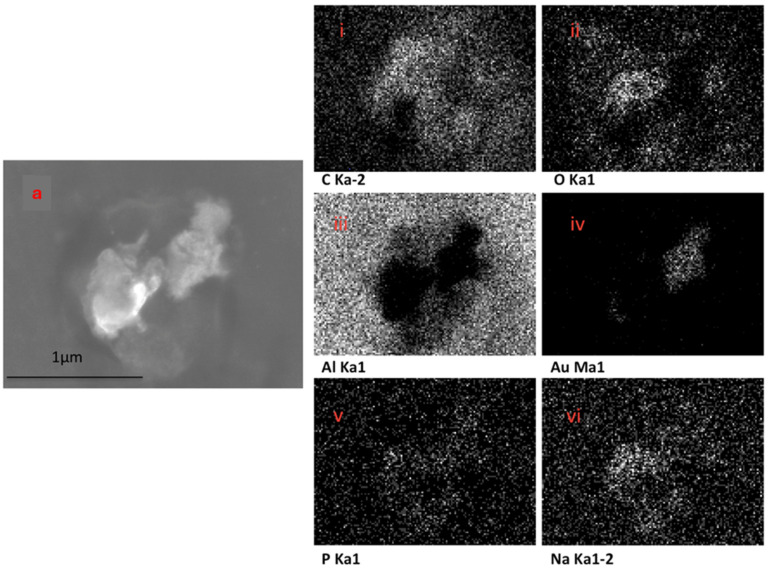
SEM backscatter of the disrupted TF–Au on aluminium foil (**a**) and EDX mapping of this TF–Au (**i**–**vi**) which shows an elemental sign of Au-NPs.

**Figure 8 pharmaceutics-16-00057-f008:**
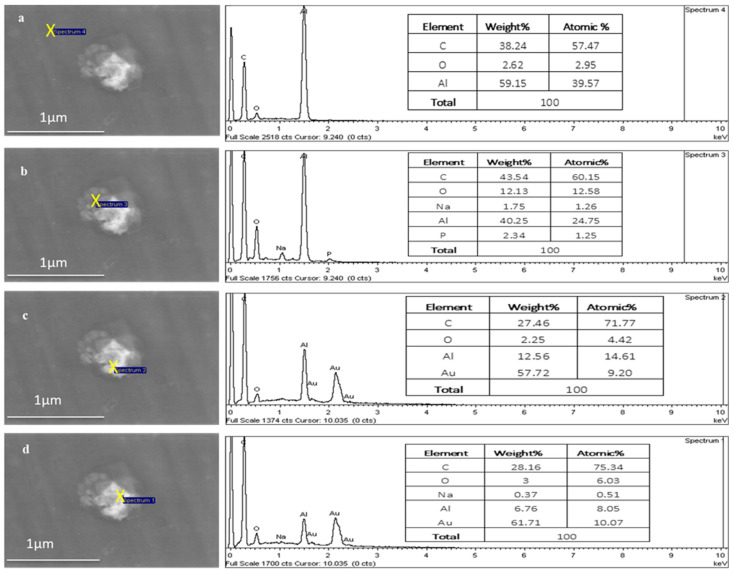
EDX of a TF–Au electrospun on aluminium foil. The Au signal increases from 0% (**a**,**b**) to 9.20% (**c**) and 10.07% (**d**) atomic percentage.

**Figure 9 pharmaceutics-16-00057-f009:**
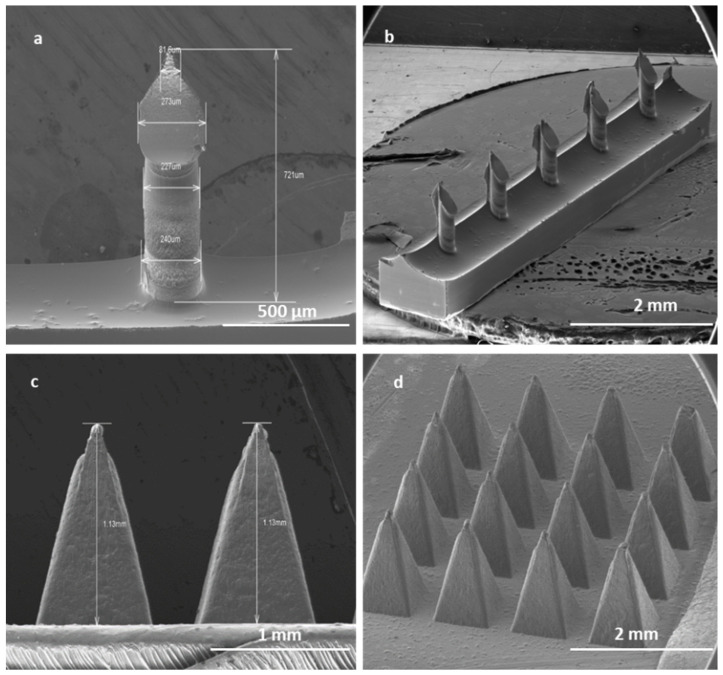
SEM image of a single SMN (**a**); array of SMNs (**b**); two single PMNs (**c**); and a patch of PMNs (**d**). Solid SMNs have an average height of 700 µm and a pitch size of 1000 µm. Solid PMNs have average height of 1 mm and an average pitch size of 300 µm.

**Figure 10 pharmaceutics-16-00057-f010:**
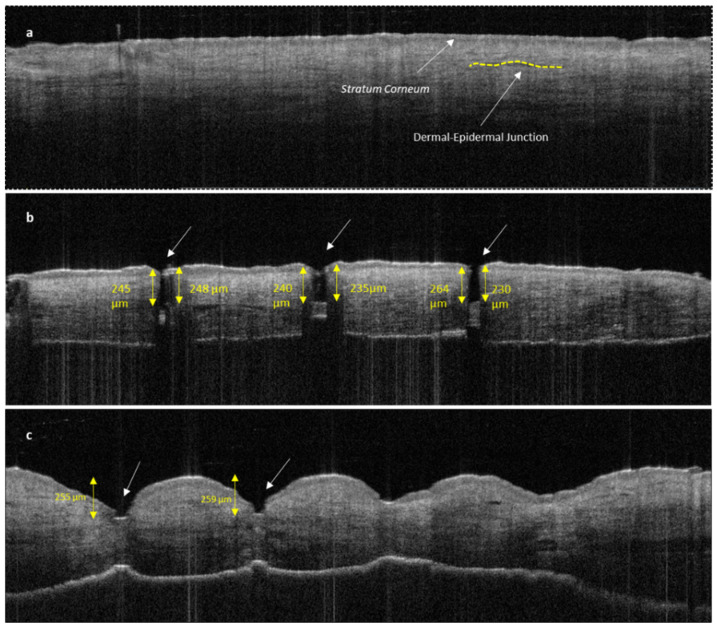
OCT image of untreated human skin (**a**); human skin treated with SMNs (**b**); and human skin treated with PMNs (**c**). Both images (**b**,**c**) show clear *stratum corneum* disruption. The length of MN insertions is approximately 230–250 μm for both SMNs and PMNs. PMNs caused significantly more indentation and skin deformation than SMNs. The white arrows indicate microneedle penetration points.

**Figure 11 pharmaceutics-16-00057-f011:**
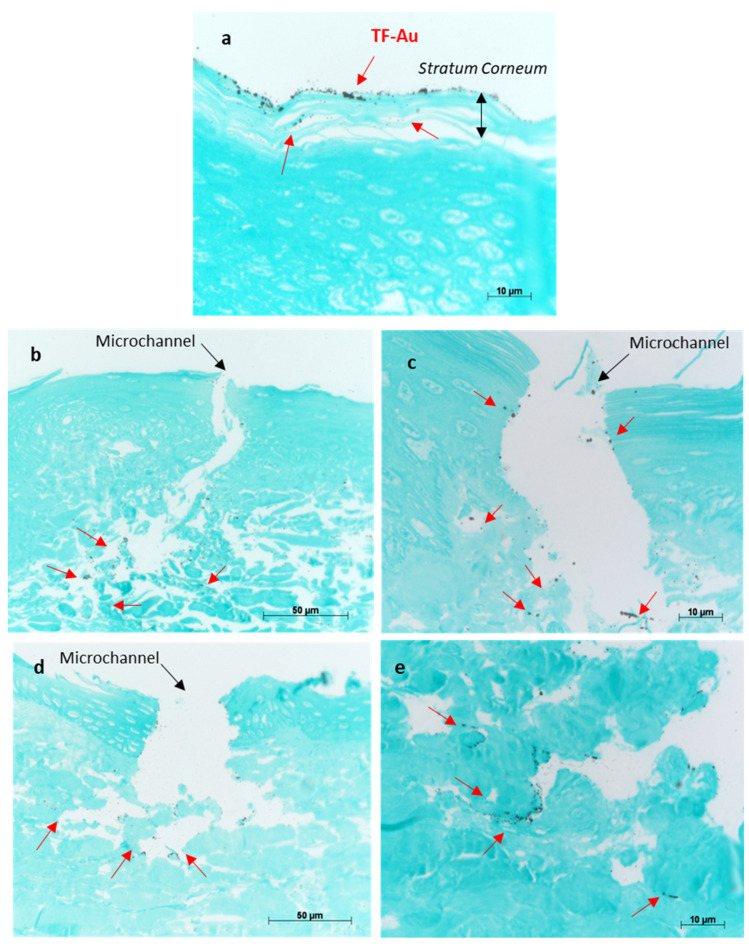
Optical images of porcine skin with applied TF–Au (**a**). Images (**b**,**c**) show different magnifications of porcine skin tissue treated with SMN+TF–Au. Images (**d**,**e**) show different magnifications of porcine skin treated with PMN+TF–Au. Red arrows indicate that the TF–Au penetrated the deeper layers of the skin (dermis).

**Figure 12 pharmaceutics-16-00057-f012:**
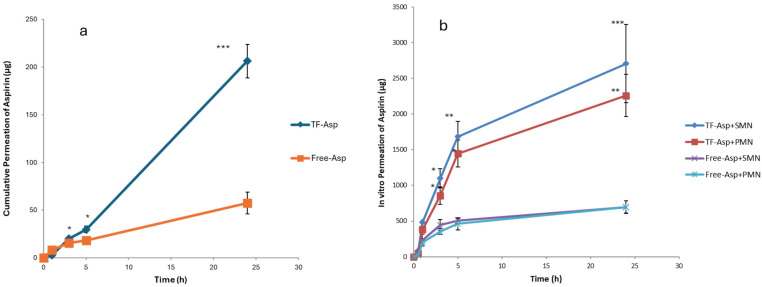
Cumulative permeation of TF–Asp and Free-Asp transported across porcine skin (**a**) compared with the cumulative permeation of TF–Asp and Free-Asp transported across porcine skin perforated with microneedles (SMNs and PMNs) (**b**). *p* ≤ 0.5, *p* ≤ 0.005, and *p* ≤ 0.001 are represented by (*), (**) and (***), respectively. N = 9.

**Figure 13 pharmaceutics-16-00057-f013:**
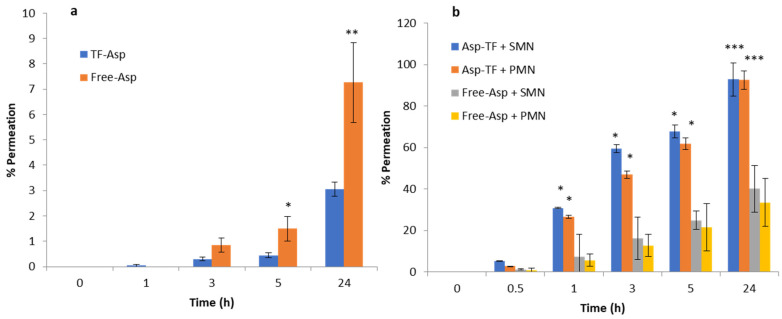
Percentage permeation of TF–Asp and Free-Asp across porcine skin (**a**) compared with the percentage permeation of microneedle-assisted (SMN and PMN) permeation of TF–Asp and Free-Asp (**b**). *p* ≤ 0.5, *p* ≤ 0.005, and *p* ≤ 0.001 are represented by (*), (**) and (***), respectively. N = 9.

**Figure 14 pharmaceutics-16-00057-f014:**
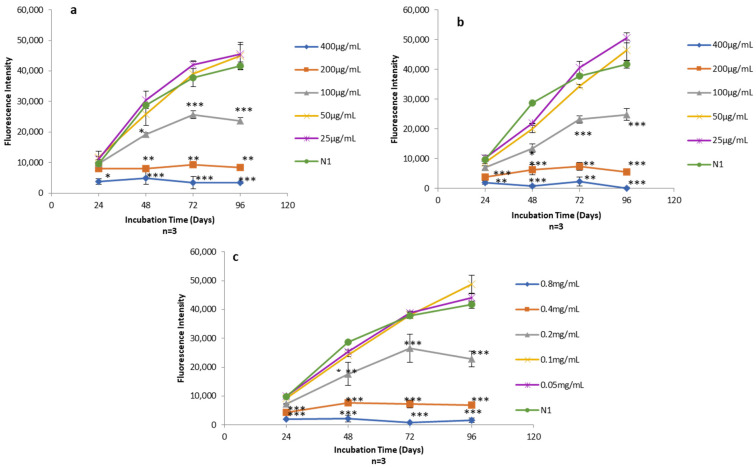
(**a**) Alamar blue assay on human fibroblasts exposed to different concentrations of Free-Asp (400 μg/mL, 200 μg/mL, 100 μg/mL, 50 μg/mL, and 25 μg/mL) compared with untreated cells (N1); (**b**) Alamar blue assay on human fibroblasts exposed to different concentrations of TF–Asp (400 μg/mL, 200 μg/mL, 100 μg/mL, 50 μg/mL and 25 μg/mL) compared with N1; (**c**) Alamar blue assay on human fibroblasts exposed to different concentrations of TFs (phospholipid concentrations of 0.8 mg/mL, 0.4 mg/mL, 0.2 mg/mL, 0.1 mg/mL and 0.05 mg/mL) compared with N1. *p* ≤ 0.5, *p* ≤ 0.005, and *p* ≤ 0.001 are represented by (*), (**) and (***), respectively. N = 3.

**Figure 15 pharmaceutics-16-00057-f015:**
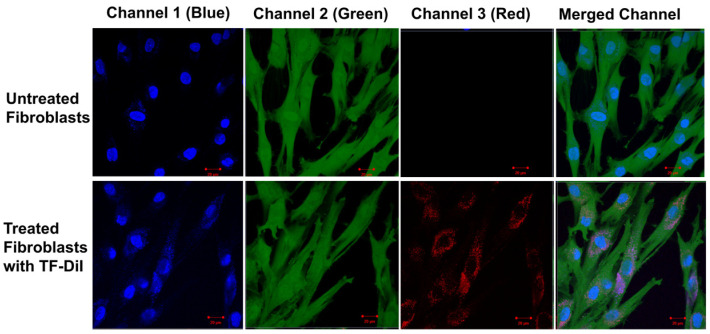
Confocal images of untreated skin fibroblasts (**top**) and treated skin fibroblasts with TF–DiI (**bottom**) after 24 h incubation time. The images show the accumulation of TF–DiI (red intensity) mainly in cell cytoplasm.

**Figure 16 pharmaceutics-16-00057-f016:**
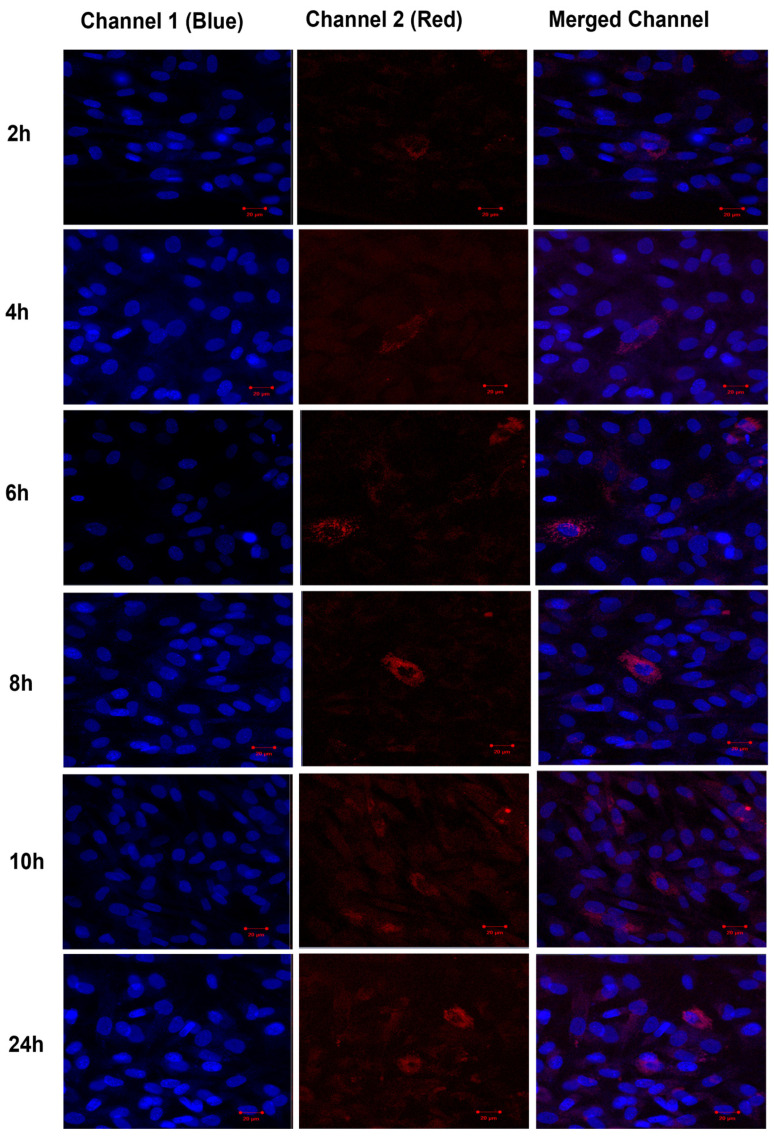
Confocal images of human fibroblasts (blue intensity, cell nuclei) treated with TF–DiI (red intensity). The images show the uptake of TF–DiI by fibroblasts over 24 h.

**Table 1 pharmaceutics-16-00057-t001:** Concentrations of aspirin and TF–Asp (used for cytotoxicity test).

Samples	Concentration (µg/mL)				
Free-Asp	400	200	100	50	25
TF–Asp	400	200	100	50	25

**Table 2 pharmaceutics-16-00057-t002:** Phospholipid concentrations of TFs used for the cytotoxicity test.

Sample	Concentration (mg/mL)				
TF	0.8	0.4	0.2	0.1	0.05

**Table 3 pharmaceutics-16-00057-t003:** The effect of different storage temperatures (4 °C, 25 °C and 37 °C) on the size and PDI of TFs. The PDI and size of TFs increase when the TFs are stored at 25 °C and 37 °C for 30 days. N = 3.

TF	Baseline	4 °C	25 °C	37 °C
Size	88.06 ± 8	89 ± 1.5 nm	182 ± 3 nm	247 ± 4 nm
PDI	0.23 ± 0.07	0.226 ± 0.07	0.239 ± 0.09	0.589 ± 0.1

**Table 4 pharmaceutics-16-00057-t004:** The size, polydispersity index (PDI) and zeta potential of TFs and TF–Asp. N = 3.

Sample	Size (nm)	PDI	Zeta Potential (mV)
TF	88.06 ± 8	0.23 ± 0.07	2.64 ± 0.41
TF–Asp	106 ± 23	0.24 ± 0.01	9.96 ± 0.4

## Data Availability

The data presented in this study are available in this article.

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
