# Peer review of "Microneedle-Assisted Transfersomes as a Transdermal Delivery System for Aspirin"

_pharmaceutics, 2023, doi:10.3390/pharmaceutics16010057_

Round 1

Reviewer 1 Report

Comments and Suggestions for Authors

Dear Authors,

                      This is well performed project, While developing a new technology we should have to think about the cost effective parameters as well. 

1.Please justify, interest of pharmaceutical industries to opt this technology in "Discussion".

2. Also justify the present technology in term of patient compliance specially keeping cost a major factor.

Comments on the Quality of English Language

can be improved but ok for publishing

Author Response

Dear Reviewer 

Many thanks for your comments. Please find the responses in the file attached.  

Kind regards

Reviewer 2 Report

Comments and Suggestions for Authors

In this article entitled (Microneedle Assisted Transferosomes as a Transdermal Delivery Vehicle for Aspirin), the manuscript is interesting however, some aspects should be better explored and explained.

Comments

Title

·        Its better to replace the word vehicle by system.

Introduction

·        Line 55, please correct minimise to be minimize.

·        Lines 64-65, the sentence was confusing and need rewrite.

·        Line 95, please replace the word recognised br recognized.

·        Line 119, please replace the word utilised by utilized.

·        Line 132, the authors used MN as abbreviation for microneedles, the should be added at first mention for microneedles.

·        Line 135, please complete the last word in the sentence (system).

·        Line 158, The word optimised Should be corrected to optimized.

Material and methods

·        Lines 190-191, this sentence was confusing (Encapsulation of drug models Au nanoparticles (Au-NPs) (provided by school of Engineering Swansea University) and DiI labelling dye, were performed using the same procedure used for aspirin encapsulation.).

·        The sentence in line 195, needs to be rewritten

·        Line 212, It is better to write (Deionized water) instead of DI water

·        Lines 218-221, please remove the bold.

·        What is the role of Au nanoparticles in this study?

·        Section 2.2.5. Energy-Dispersive X-Ray Spectroscopy (EDX), the authore mention SEM. What is the relation?

·        How many formulations of asprin transferosomed did the authors develop?

Results and discussion

·        Result and discussion was clear and well witten.

Conclusion

·        The conclusion is well written

References

·        The authors didn’t follow the author's guidelines regarding the references. They should use MDPI reference style. I recommend Zotero software to arrange the references 

Comments on the Quality of English Language

Moderate editing of English language required

Author Response

(The authors gave the same response as above.)

Reviewer 3 Report

Comments and Suggestions for Authors

The paper is well done and a good novelty in transdermal delivery. It May be accepted with minor revision

 Line 181: remove one of the round brackets

 Line 316: The temperature of the Franz Cells was set and maintained at 32°C, normally it’s 37°C because it must mimic blood temperature, why didn’t you use that temperature? Explain why you used a lower temperature

Author Response

(The authors gave the same response as above.)

Reviewer 4 Report

Comments and Suggestions for Authors

The study investigated the effect of combining transferosomes and microneedle penetration on transdermal delivery of aspirin, with potential therapeutic effects in various dermatological disorders, including melanoma.

The abstract is appropriate for the content of the text, but the authors should:

- replace the term “applications” (lines 15-16) with “administrations”, which I consider is more suitable in this case;

- reformulate the phrase regarding the OCT test, which according to the described methodology, was performed only on human skin, not for perforated porcine skin.  

The article is well constructed and clearly written and the experiments were well conducted by appropriate methodology. The methods are clearly described, and the results are supported by enough references and statistical analysis.

The obtained results are sustained by tables and numerous figures.  

The conclusions are consistent with the evidence and arguments presented through the manuscript and they do address the main question posed.

However, some items should be addressed by the authors before publication.

-          The authors should explain the reason for using human skin only for OCT test and the porcine skin for the other permeation studies.

-          Also, the authors should provide more information about the body region of the animals from which they dermatomed the skin samples.  

-          In the permeation studies, please clarify in which form (formulation) were TF, TF-Au, and TF-Asp applied/used.

-          To achieve the intended therapeutic effect, on what basis was the aspirin concentration selected?

Author Response

Dear Reviewer

Many thanks for your comments. Please find the responses in the attached file.

Kind regards

Reviewer 5 Report

Comments and Suggestions for Authors

Introduction: in the part that concerns the release systems of poorly soluble drugs to facilitate their solubilization and skin penetration, nanostructures/nanomicelles also deserve to be mentioned, there are some recent examples (10.3390/molecules24091793; 10.3390/pharmaceutics12111078). The part dedicated to transferosomes is too large, we are not in the context of a review. It must be significantly reduced with a consequent reduction in the number of bibliographic references, which are redundant.

Materials and methods - In vito release of aspiin using...: specify the release medium

Results - Drug release: the authors should better explain/comment the results of the release tests. Drug amount released over time by TFs seems very low, is it enough to be effective? Are we certain that the drug comes out over time or is that the maximum amount of drug that can be released from the transferosomes?

Fig. 12 - What does skin background refer to? Please explain better and perhaps it is not necessary to report it in the graph. What does skin background refer to? Please explain better and perhaps it is not necessary to report it in the graph. Rewrite the figure caption more clearly

Author Response

(The authors gave the same response as above.)

Round 2

Reviewer 1 Report

Comments and Suggestions for Authors

Dear Authors,

                     The revision provides adequate improvement in the manuscript.

Reviewer 2 Report

Comments and Suggestions for Authors

Authors addressed all required comments

Reviewer 5 Report

Comments and Suggestions for Authors

accepted in this form